# Effects of Continuous Glucose Monitoring on Glycemic Control in Type 2 Diabetes: A Systematic Review and Meta-Analysis

**DOI:** 10.3390/healthcare12050571

**Published:** 2024-02-29

**Authors:** Seung-Yeon Kong, Mi-Kyoung Cho

**Affiliations:** 1Referral Center, Chungbuk National University Hospital, Cheongju 28644, Republic of Korea; rrhd2@cbnuh.or.kr; 2Department of Nursing Science, Chungbuk National University, Cheongju 28644, Republic of Korea

**Keywords:** type 2 diabetes, continuous glucose monitoring, glycemic control

## Abstract

As the prevalence of diabetes is rapidly increasing, the use of continuous glucose monitoring, which is effective in improving glycemic control in type 2 diabetes, is increasing. Methods: Systematic review was performed according to PRISMA criteria. The search was conducted for articles published until 31 May 2023 in PubMed, CINAHL, Cochrane Library, EMBASE, ClinicalKey, etc. The meta-analysis involved the synthesis of effect size; tests of homogeneity and heterogeneity; trim and fill plot; Egger’s regression test; and Begg’s test for assessing publication bias. Results: 491 studies were searched, of which 17 studies that met the selection criteria were analyzed. The overall effect on HbA1c was −0.37 (95% CI, −0.63~−0.11, *p* < 0.001), with HbA1c decreasing significantly after CGM interventions. Sub-analyses showed that the study was statistically significant in those aged 60 years or older, when rt-CGM was used and when the study was performed in multiple centers. Conclusion: The results of this study showed that intervention using CGM was effective in reducing HbA1c in type 2 diabetes. The factors identified in this study can be used as guidelines for developing future CGM intervention programs.

## 1. Introduction

Diabetes is an insidious, chronic disease, and its incidence is increasing rapidly worldwide. As indicated by the Korean Diabetes Fact Sheet, the number of diabetic patients in South Korea aged 30 or more reached around 6 million in 2020 and showed a consistent upward trend [1]. However, only 24.5% of patients had successfully managed their diabetes, as determined by the key indicator, HbA1c, which should ideally be below 6.5% [2].

A recent analysis conducted by Kaptoge et al. (2023) elucidated the significant impact of an early diabetes diagnosis on life expectancy, revealing a marked reduction of approximately 3 to 4 years for every decade of life. The study notably highlighted that the earliest age of diabetes diagnosis was predominantly associated with an increased prevalence of vascular diseases, including myocardial infarction and stroke, alongside other non-neoplastic causes of death, such as respiratory, neurological, and infectious diseases, and external causes [3]. Furthermore, it was found that approximately 28.6% of individuals with early-diagnosed diabetes develop major vascular complications, including cardiovascular, cerebrovascular, or peripheral artery diseases. Conversely, a substantial 67.2% of individuals encounter microvascular complications, notably retinopathy, nephropathy, or neuropathy [4]. Given that the prognoses of diabetic patients depend heavily on the presence of complications, the prevention of chronic diabetic complications by the diligent self-management of glycemic control is the foremost priority [5,6]. 

Self-monitoring of blood glucose (SMBG) is widely accepted to be the most effective means of achieving long-term blood sugar control in diabetic patients [7]. However, despite its effectiveness, SMBG is limited by its invasive nature and associated pain and inconvenience, which leads to reduced patient compliance, especially when patients are accompanied by others [8]. Furthermore, SMBG results provide limited understanding of specific blood glucose fluctuations, such as postprandial glucose spikes or asymptomatic hypoglycemia [9]. Consequently, continuous glucose monitoring systems (CGMs) have been increasingly utilized to address these limitations by providing real-time blood glucose readings to patients. In addition, CGM information can positively impact treatment planning, medication regimens, self-blood glucose monitoring schedules, and the adoption of appropriate lifestyle habits. In particular, CGM data are extremely useful for establishing more accurate diagnosis and treatment plans and enabling blood glycemic control [10]. 

Studies have demonstrated that CGM use leads to improved self-management behaviors, enhanced blood glycemic control, effective reductions in HbA1c levels, and hypoglycemic improvements in diabetic patients [11,12,13,14]. For these reasons, CGMs have been increasingly used, even by type 2 diabetic patients. However, the majority of investigative studies on the effects of CGM have focused on type 1 diabetes, and its effects on type 2 diabetes have received relatively little attention. Nevertheless, a recent large-scale retrospective cohort study on CGM reported that patients with type 2 diabetes taking insulin showed greater HbA1c improvements than patients with type 1 diabetes [15]. This observation suggests that CGM may be more effective in type 2 diabetic patients and prompts questions regarding whether the levels of glycemic control provided by CGM and SMBG differ in type 2 diabetic patients. Meta-analyses of CGM in type 2 diabetic patients conducted to date have only included a limited number of randomized controlled trials (RCTs) and primarily focused on the impact of CGM intervention on HbA1c levels without investigating whether CGM directly ameliorates hypoglycemia or influences psychological or physiological factors, such as weight, BMI, or cholesterol. 

In this study, we aimed to enhance the evidence base for CGM interventions in type 2 diabetic patients by comprehensively evaluating the effects of interventions on glycemic control and physiological and psychological factors and providing a substantiated rationale for the use of CGM as an effective intervention in type 2 diabetic patients. 

This study systematically reviews the characteristics and key findings of studies that validated the effectiveness of intervention programs utilizing CGM in type 2 diabetic patients. 

## 2. Materials and Methods

### 2.1. Study Design

Systematic literature review and meta-analysis were utilized to analyze the impact of CGM intervention on glycemic control in type 2 diabetic patients. 

### 2.2. Inclusion and Exclusion Criteria

The reviewed literature was analyzed according to the Preferred Reporting Items for Systematic Reviews and Meta-analysis (PRISMA) [16]. The PRISMA 2020 Checklist is presented as Appendix A. A systematic literature search based on PICO-SD (participants, intervention, comparison, outcomes, study design) was conducted to select literature for analysis. Participants (P) were type 2 diabetes patients aged over 18. Intervention (I) using a continuous glucose monitoring (CGM) were included, regardless of the CGM type. The control group (C) consisted of patients that received usual care and self-monitoring of blood glucose (SMBG). The outcome was glycemic control. Only randomized controlled trials (RCTs) were included to ensure objective evidence on intervention effectiveness. All studies compared two groups with HbA1c as the outcome variable and provided convertible statistical data (sample sizes, means, standard deviations, and effect sizes). Studies were published in Korean or English before 31 May 2023. Studies or theses not available as original text, survey research, and single-group comparative studies were excluded. 

### 2.3. Literature Search Strategy

The literature search was conducted based on COSI (COre Standard, Ideal) provided by the National Library of Medicine (NLM) using the following core databases [17]: international databases such as PubMed, CINAHL, Cochrane Library, EMBASE, and ClinicalKey and domestic databases such as Research Information Sharing Service [RISS], KMbase, KISS, and KoreaMed. The search was conducted for studies published up to 31 May 2023. The primary search terms used were ‘diabetes mellitus, type 2’ [MeSH Terms], ‘continuous glucose monitoring’, and ‘glycemic control’ [MeSH Terms]. For domestic databases, the search was conducted using combinations of type 2 diabetes, continuous glucose monitoring, and glycemic control. In addition, manual searches were conducted for studies included as references, and the Google Scholar search engine was utilized for related research topics. This review protocol was registered with Prospero registration no. CRD42024505351 available at https://www.crd.york.ac.uk/prospero/#recordDetails (accessed on 3 February 2024).

### 2.4. Quality Assessment of the Selected Studies

The quality of selected studies was assessed using the checklist for RCT studies included in Joanna Briggs Institute of Critical Appraisal Tools [18]. This checklist comprises 13 items and are as follows: random assignment, allocation concealment, treatment group similarity, blinding of participants, blinding of delivered treatment, blinding of outcome assessor, similar treatment, follow-up completion, intention-to-treat analysis, consistent method of assessing outcome measures in groups, reliability of outcome measures, appropriate statistical analysis, and appropriate trial design. Each item received a score of 0 (‘no’ or ‘unclear’) or 1 (‘yes’), and thus the maximum possible score was 13 points. One reviewer performed this assessment for each study, and a second reviewer confirmed the results. Discrepancies were resolved by discussion and consensus.

### 2.5. Selection Process for the Analyzed Literature

Two researchers independently reviewed the identified studies. A list of identified studies from domestic and international databases was compiled using Microsoft Excel 2016, and duplicate studies were removed. Subsequently, titles and abstracts were reviewed to determine whether studies met the selection criteria. Finally, full texts were reviewed, and studies were selected for analysis. 

### 2.6. Data Coding

Author names, publication years, countries, number of research centers, funding, participant numbers and characteristics, study design, type of CGM, intervention period, comparator, outcomes, and quality assessment scores were recorded. The Libre had ‘flash’ CGM (fCGM) as the sensor and had to be scanned at least every 8 h to download the data to the reader. However, We coded Libre’s CGM as real-time CGM in this study because participants could receive results immediately without waiting for a doctor. The following outcome variables were subjected to effect size analysis: HbA1c, weight, BMI, SBP/DBP, hypoglycemia, hyperglycemia, and time in range, average blood glucose level, distress, QoL, satisfaction, and HDL-cholesterol. Two researchers conducted data coding independently, and disagreements were resolved by consensus based on the joint reviews of original texts.

### 2.7. Data Analysis

Data analysis was performed using MIX 2.0 Pro, version 2.015 (MIX Professional software for meta-analysis in Excel) [19]. For all study outcomes, Hedge’s g was utilized as the effect size, considering that many studies had a small sample size [20]. Hedge’s g values were interpreted as follows: an effect size of ≥0.2 but <0.5 was categorized as small, an effect size of ≥0.5 but <0.8 as medium, and an effect size of ≥0.8 as large [21]. The significance level for effect size was set at 0.05, and the confidence interval (CI) at 95%. The analysis was conducted using a random effects model because of the variances exhibited by study participants and study heterogeneity. Heterogeneity was assessed using the I-squared (I^2^) statistic and was deemed absent when I^2^ was 0%, medium at 50%, and high at 75% [22]. Egger’s regression and Begg’s tests and the trim and fill method were used to confirm publication bias [23,24].

## 3. Results

### 3.1. Literature Selection

Overall, 491 studies were identified during the initial search, but only 7 were included after applying study selection and exclusion criteria. However, 10 additional studies were selected by reviewing references in these 7 papers and performing a search using the Google Scholar search engine. Thus, 17 studies were included in the analysis (Figure 1). 

### 3.2. Characteristics of the Included Studies

A total of 10 of the 17 studies included were published after 2015, and higher number of studies were conducted in the United States than in other countries (6 of the 17). Twelve studies were conducted across multiple centers. All 17 studies were funded, and all were RCTs. In total, 1619 patients were involved. The CGM devices used for interventions were 11 real-time CGM and 6 retrospective CGM. The most common intervention period was 12 weeks; six studies adopted this timeframe. In control groups, SMBG was performed in the normal manner. In all studies, HbA1c was used as the outcome variable. In 8 studies, CGM data and physiological variables were measured as follows: weight in 5 studies, BMI in 4, BP in 4, and HDL-cholesterol in 2. Distress and satisfaction were assessed in three studies and QoL in four (Table 1). 

### 3.3. Quality Assessment

The average quality assessment score for the 17 studies was 8 points (range: 6–9). All 17 had a suitable RCT design and clearly described the random assignment procedure used. Participants were not blinded in any study, and information on assessor measurement reliability was not provided; thus, it was assessed as unclear. The mediator or measurer was blinded in one study apiece (Table 2).

### 3.4. Effect of CGM Intervention on HbA1c

Overall, CGM intervention significantly decreased HbA1c, as indicated by Hedge’s g = −0.37 (95% confidence interval [CI]: −0.63, −0.11, *p* < 0.001) (Figure 2). The I^2^ was 82.7% (Q = 92.35, Q−df = 74.35, *p* < 0.001), indicating a high level of heterogeneity, thereby suggesting a need for exploratory explanations of the heterogeneity in effect sizes (Figure 2). Sub-analysis was conducted based on study characteristics, such as country, number of participants, number of centers, CGM intervention types, intervention period, quality assessment scores, and insulin therapy. Studies that targeted participants aged 60 or older, studies conducted at multiple centers, studies utilizing real-time CGM for interventions, and studies with reported quality assessment scores of ≤8 (Table 3).

Meta-regression analysis was also conducted to investigate heterogeneity potentially arising from differences between studies and participants. The moderators used in the meta-regression analysis to explain heterogeneity were country, number of participants, number of research centers, CGM type, intervention period, quality assessment scores, and insulin therapy. A significant reduction in HbA1c was observed in studies that enrolled participants ≥ 60 (Z = −2.06, *p* = 0.039), studies using real-time CGM (Z = −4.45, *p* < 0.001), studies with quality assessment scores of ≤8 (Z = −4.15, *p* < 0.001), and studies receiving insulin therapy (Z = −2.49, *p* = 0.013) (Table 4).

### 3.5. Effect of CGM Intervention on Secondary Outcomes

In addition to HbA1c (the primary outcome variable), various secondary outcomes such as CGM data, physiological factors (weight, BMI, cholesterol), and psychological factors (distress, satisfaction, and quality of life (QoL)) were also measured. However, results showed that CGM intervention had no overall effect on secondary outcomes (Table 5).

### 3.6. Publication Bias Analysis

A funnel plot was used to verify the validity of the analyzed results and to assess publication bias. The plot showed that effect sizes were not symmetrically distributed around the central dotted line (Figure 3). Egger’s regression test, Begg’s test, and the trim and fill method were used to determine whether the degree of asymmetry was significant. The results indicated no publication bias (Table 6). Trim and fill analysis showed that the original combined effect size of CGM intervention was −0.36, and the adjusted overall effect size was −0.58 (95% CI: −0.83, −0.33), which resulted in an effect size increase from a small to an intermediate level (Table 6). Furthermore, when the six studies indicated by white circles in the plot were added to the left of the filled synthetic line, it appeared that publication bias had been corrected (Figure 4).

## 4. Discussion

Strategies designed to improve blood glucose management in diabetic patients are attracting considerable interest because of the rapidly increasing prevalence of diabetes. In particular, CGM are now viewed as crucial for blood glucose measurements, and their use is increasing among type 2 diabetes patients [40]. This study aimed to systematically review the characteristics of CGM interventions conducted in type 2 diabetic patients and integrate their effects to provide a foundation for developing effective CGM interventions. 

All 17 RCT studies included in the meta-analysis were performed on patients over 18 years old. These studies were conducted in various regions, including the United States, Europe, and Asia. Most studies were conducted across multiple centers, and the predominant study period was 12 weeks. The control group received normal diabetes care based on SMBG. HbA1c was used as the primary effect variable, and physiological factors (CGM data, weight, BMI, BP, cholesterol) and psychological factors (distress, satisfaction, and QoL) were used as secondary effect variables. 

The quality assessment revealed that participant allocation was random in all studies. However, the blinding of participants, interveners, and assessors was not achieved, and there was insufficient information regarding the reliability of assessor measurements, which resulted in an unclear rating. This lack of information is believed to be due to the inherent bias associated with studies involving the insertion of CGM devices into patients’ bodies. Future research should focus on devising methods to minimize potential biases related to the blinding of researchers, interveners, and assessors to facilitate the more rigorous evaluation of the effects of CGM. 

In this study, the primary effect variable for glycemic control was HbA1c, and the overall effect of CGM intervention on HbA1c was −0.37, indicating a significant reduction in HbA1c levels. This result was greater than the −0.25 reported in a meta-analysis conducted by Janapala et al. (2019) [41] and similar to the −0.35 reported by Ida et al. [42]. The intervention methods used in this study were categorized as retrospective or real-time CGM. The patients who underwent retrospective CGM analyzed their blood glucose patterns retrospectively, unaware of their results at time of measurement, whereas those who underwent real-time CGM viewed CGM data in real-time [40]; eleven studies used real-time CGM intervention and six used retrospective CGM intervention. Meta-analysis showed real-time CGM intervention reduced HbA1c, which concurs with the findings of Ida et al. [42] and suggests that access to real-time blood glucose values and trends enables patients to reduce hyperglycemia, extend time in target blood glucose ranges, and prevent hypoglycemia [43,44]. These findings suggest that real-time CGM-based interventions are more effective at reducing HbA1c levels in type 2 diabetes patients than retrospective CGM-based interventions. 

Furthermore, HbA1c reductions were more pronounced in individuals aged ≥60, which aligns with the findings of previous domestic and international studies [45,46] and indicates that younger age is associated with more difficult blood glycemic control. Individuals aged under 60 tend to be office workers with self-management challenges due to factors such as sleep deprivation caused by work-related stress, lack of exercise due to long working hours, and a social drinking culture [47]. Therefore, there is a need to develop an efficient way to manage blood sugar levels using CGM, which can be used by office workers without restrictions on time and place.

Finally, meta-regression analysis was conducted by entering study and participant characteristics to identify the sources of systematic heterogeneity across studies. This analysis showed that the following moderators explained heterogeneity, i.e., country, number of participants, number of research centers, CGM type, intervention period, quality assessment scores, and insulin therapy, which cautions that these characteristics might influence the heterogeneity of results. The results of the meta-regression analysis of this study showed that CGM was effective in reducing HbA1c when the intervention period was 24 weeks or more compared to when the intervention period was less than 24 weeks. These results show that studies with relatively long intervention periods are effective in reducing glycated hemoglobin [48]. However, the study by Furler et al. (2020) [29], which was conducted for 52 weeks, the longest period in this study, showed no HbA1c reduction effect, and this result was due to the use of a professional-mode flash glucose monitoring sensor device that did not allow patients to check glucose data. Therefore, to determine the intervention effect of diabetes management on changes in glycated hemoglobin in future long-term studies, the primary outcome should be measured at 12 weeks using an rt-CGM device to confirm changes in HbA1c and confirm whether the effect lasts for more than 24 weeks.

## 5. Conclusions and Recommendations

In this study, we aimed to elucidate the effects of CGM interventions on glycemic control in type 2 diabetes patients by meta-analysis. Our goal was to improve the level of evidence for intervention methods and offer suggestions for future research. A total of 17 RCT studies were analyzed, and CGM intervention was found to diminish HbA1c levels. Furthermore, real-time CGM effectively reduced HbA1c levels at multiple centers when an Intervention period of >24 weeks was used. The factors identified in this study might serve as basic guidelines for setting the direction of future CGM intervention research and developing and implementing effective intervention programs. Unfortunately, most of the studies included did not provide details of education during intervention. In previous studies, when no systematic education was provided during CGM usage, no HbA1c decrease was observed [48]. Improved glycemic control was only evident when specialized education was provided [30]. Thus, we suggest that studies should be undertaken to confirm the effects reported by studies conducted with a systematic education program. The limitations of this study include the possibility that some relevant literature may have been omitted during the literature search. Furthermore, caution is required when interpreting our results due to the presence of heterogeneity. Our analysis did not encompass the rate of hypoglycemia or the time spent in the glycemic target range, which are important outcomes for evaluating the effectiveness and safety of CGM interventions. We suggest that future research should include these various variables to provide a more holistic view of CGM’s impact on diabetes management. The inclusion criteria for the literature in our study were to merge the first measured value after the end of the program. In future research, we suggest an analysis that subdivides the types of CGM in glycemic control and considers the long-term effects and frequency of application of CGM. Additionally, we suggest that future studies should be undertaken to validate the effectiveness of CGM, focusing on devising methods that minimize these limitations. 

## Figures and Tables

**Figure 1 healthcare-12-00571-f001:**
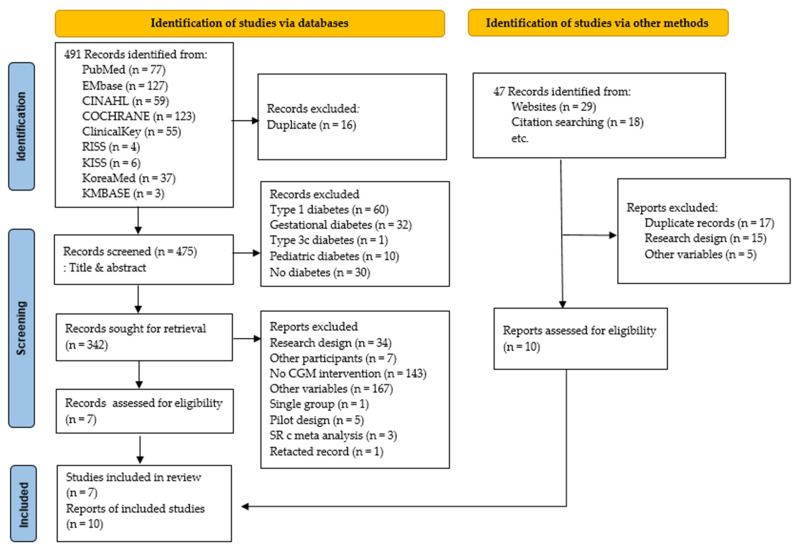
PRISMA flow diagram.

**Figure 2 healthcare-12-00571-f002:**
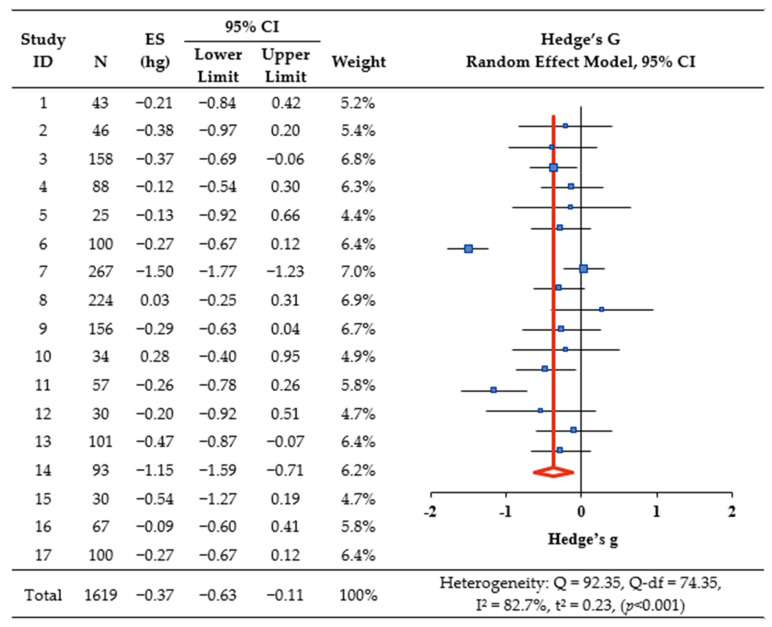
The effect of CGM on HbA1c. Notes. ES: effect size; CI: confidence interval.

**Figure 3 healthcare-12-00571-f003:**
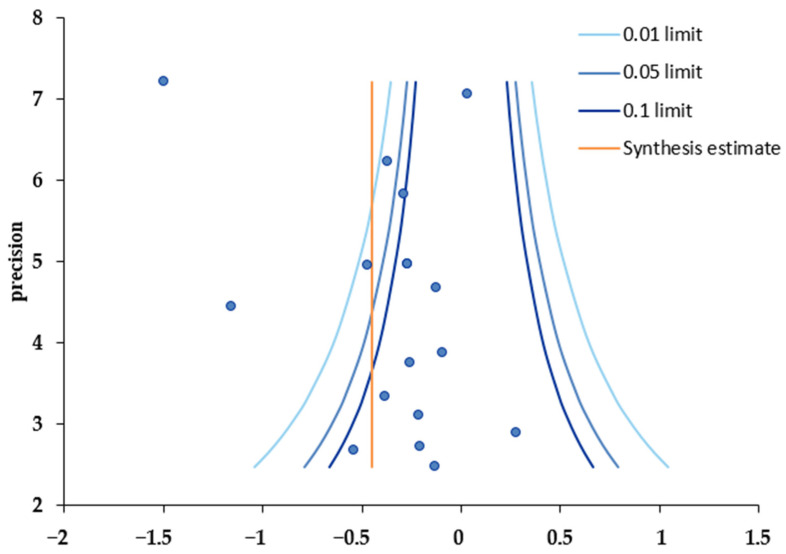
Funnel plot.

**Figure 4 healthcare-12-00571-f004:**
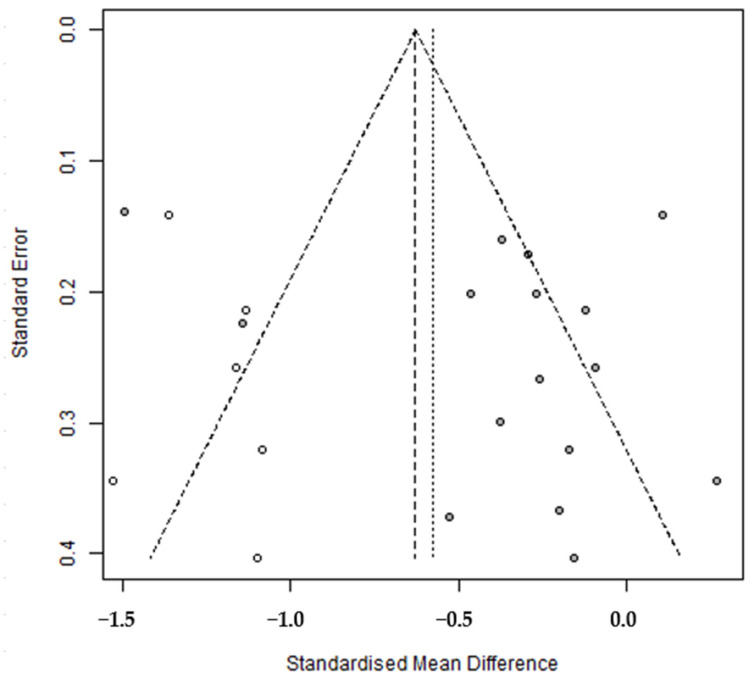
Trim and fill.

**Table 1 healthcare-12-00571-t001:** Characteristics of the included studies.

Study ID	Author	Year	Country	Center	Fund	Participants	Characteristics of Participants	Type of CGM	Intervention Period (Weeks)	Comparator	Outcome Variables
1	Ajjan et al. [25]	2016	UK	9	Yes	N = 45 (E:30, C:15)	Age ≥ 18 years HbA1c 7.5–12.0% Receiving insulin therapy > 6 months	FreeStyle Navigator (Abbott, Chicago, IL, USA)	25	SMBG	CGM data HbA1c Body weight (kg) Blood glucose testing Frequency (tests/day)
2	Allen et al. [12]	2008	US	2	Yes	N = 46 (E:21, C:25)	Age > 20 years HbA1c > 7.5% Physical activity ≤2 days/week Not receiving insulin therapy	Minimed (Medtronic, Northridge, CA, USA)	8	SMBG	HbA1c Physical activity Self-efficacy BP, BMI
3	Beck et al. [14]	2017	US	25	Yes	N = 158 (E:79, C:79)	Age > 25 years HbA1c 7.5–10.0% Receiving insulin therapy > 1 year Stable medication regimen and weight >3 months SMBG ≥ 2/day Estimated glomerular filtration rate > 45 mL/min/1.73 m^2^	Dexcom G4 Platinum (Dexcom, San Diego, CA, USA)	24	SMBG	HbA1c Hypoglycemia QoL
4	Blackberry et al. [26]	2014	Australia	22	Yes	N = 88 (E:46, C:42)	Age 18–80 years HbA1c ≥ 7.5% No previous experience with insulin therapy Stable OHA regimen > prior 3 months SMBG ≥ 2/day	iPro2^TM^ (Medtronic, Northridge, CA, USA)	24	SMBG	HbA1c QoL CGM satisfaction 36 Health survey questionnaire version 2 (SF-36 v2)
5	Cosson et al. [27]	2009	France	5	Yes	N = 25 (E:11, C:14)	Age 40–70 years HbA1c 8.0–10.5% Stable OHA and insulin regimen prior to >3 months SMBG ≥ 4/week No previous experience with CGM	The GlucoDay system (Menarini Diagnostics, Florence, Italy)	12	SMBG	HbA1c Glycemic control (Changes in 48 h CGM data) Hypoglycemia
6	Ehrhardt et al. [28]	2011	US	1	Yes	N = 100 (E:50, C:50)	Age ≥ 18 years HbA1c 7.0–12.0% Diagnosis ≥ 3 months SMBG 4/day Treated with diet or exercise Not receiving prandial insulin	DexCom^TM^ SEVEN (DexCom)	12	SMBG	HbA1c Glycemic control Weight BP Stress
7	Furler et al. [29]	2020	Australia	25	Yes	N = 299 (E:149, C:150)	Age 18–80 years HbA1c ≥ 7.0% Diagnosis ≥ 1 year Receiving OHA or Insulin therapy	FreeStyle Libre Pro (Abbott)	52	SMBG	HbA1c CGM data Distress
8	Haak et al. [30]	2016	European	26	Yes	N = 224 (E:149, C:75)	Age ≥ 18 years HbA1c 7.5–12.0% Receiving insulin therapy ≥ 6 months (current regimen ≥ 3M SMBG ≥ 10/week at least 2 months	FreeStyle Libre^TM^ (Abbott)	24	SMBG	HbA1c CGM data QoL
9	Martens et al. [31]	2021	US	15	Yes	N = 156 (E:105, C:51)	Age ≥ 30 years HbA1c 7.8–11.5% Diagnosis and insulin therapy ≥ 6 months SMBG ≥ 3/week	Dexcom G6 (Dexcom)	32	SMBG	HbA1c Height Weight Cholesterol CGM satisfaction
10	Sato et al. [32]	2016	Japan	1	Yes	N = 34 (E:17, C:17)	Age > 20 years HbA1c 6.9–11.0% Receiving insulin therapy	iPro^®^ 2 (Medtronic)	32	SMBG	HbA1c Diabetes Treatment Satisfaction (DTSQ)
11	Yoo et al. [33]	2008	Korea	1	Yes	N = 57 (E:29, C:28)	Age 20–80 years HbA1c 8.0–10.0% Receiving OHA or insulin therapy ≥ 1 year Stable insulin or OHA regimen ≥ prior 2 months Stable OHA or lipid-lowering drugs ≥4 weeks	Guardian RT (Medtronic)	12	SMBG	HbA1c FBS, PP2, Lipid profiles, Weight, Waist circumference BMI, Fat consumption Cholesterol intake (g/day) Exercise time (min/week)
12	Yeoh et al. [34]	2018	Singapore	1	Yes	N = 30 (E:14, C:16)	Age ≥ 21 years HbA1c > 8% Type 2 diabetes with CKD stage 3 (eGFR 30–60 mL/min per 1.73 m^2^) Above (pre-dialysis) for >3 months Sustained for >6 months Receiving insulin and/or OHA	iPro device (Medtronic)	12	SMBG	HbA1c CGM data
13	Ajjan et al. [35]	2019	England	22	No	N = 102 (E:50, C:52)	Age ≥ 18 years HbA1c 7.5%–12.0% Receiving insulin therapy ≥ 6 month	FreeStyle Libre Pro^TM^ (Abbott)	28	SMBG	HbA1c CGM data Treatment satisfaction (DTSQ)
14	Wada et al. [36]	2020	Japan	5	Yes	N = 93 (E:48, C:45)	Age 20–70 years HbA1c 7.5–8.5%	Free Style Libre (Abbott)	24	SMBG	HbA1c Weight, BP Diabetes medication change (DTSQ)
15	Moon et al. [37]	2022	Korea	3	Yes	N = 30 (E:15, C:15)	Aged 30 to 65 years HbA1c 7.5–10.0% Receiving OHA Treated without insulin ≥ 3 months	Guardian 3 (Medtronic MiniMed, Northridge, CA, USA)	24	SMBG	HbA1c CGM data, BP Lipid variables, Weight, Satisfaction K-DMSES, ADS-K, SDSCA-K
16	Price et al. [38]	2021	US	8	Yes	N = 68 (E:45, C:23)	Age ≥ 30 years HbA1c 7.8–10.5% Treated with two or more noninsulin antidiabetic drugs Stable body weight over the past 3 months	Dexcom G6 (Dexcom)	12	SMBG	CGM data HbA1c Adverse Events
17	Vigersky et al. [39]	2012	US	1	Yes	N = 100 (E:50, C:50)	Age ≥ 18 years HbA1c 7.0–12.0% Diagnosis ≥ 3 months Not receiving prandial insulin SMGB 4/days	DexCom SEVEN (DexCom)	12	SMBG	HbA1c Weight BP Stress

Notes. E: experimental group; C: control group; CGM: continuous glucose monitoring; SMBG: self-monitoring of blood glucose; MDI: multiple daily injection; OHA: oral hypoglycemia agent; HbA1c: glycosylated hemoglobin; BP: blood pressure; and BMI: body mass index.

**Table 2 healthcare-12-00571-t002:** Quality assessment of the included studies.

Joanna Briggs Institute of Critical Appraisal Tools Checklist for Randomized Controlled Trials														
Study ID	1	2	3	4	5	6	7	8	9	10	11	12	13	Total Score
1	1	1	0	0	0	0	1	1	1	1	0	1	1	8
2	1	1	1	0	0	0	1	1	1	1	0	1	1	9
3	1	1	0	0	0	0	1	1	1	1	0	1	1	8
4	1	1	1	0	0	0	1	1	1	1	0	0	1	8
5	1	1	0	0	1	0	1	1	1	1	0	1	1	9
6	0	0	1	0	0	0	1	1	1	1	0	0	1	6
7	1	1	1	0	0	1	1	1	1	1	0	0	1	9
8	1	1	0	0	0	0	1	1	1	1	0	1	1	8
9	1	1	1	0	0	0	1	1	1	1	0	1	1	9
10	1	1	1	0	0	0	1	1	1	1	0	1	1	9
11	1	1	1	0	0	0	1	1	1	1	0	1	1	9
12	0	0	0	0	0	0	1	1	1	1	0	1	1	6
13	1	1	1	0	0	0	1	1	1	1	0	0	1	8
14	1	0	1	0	0	0	1	1	1	1	0	1	1	8
15	1	1	1	0	0	0	1	1	1	1	0	1	1	9
16	0	0	1	0	0	0	1	1	1	1	0	1	1	7
17	0	0	0	0	0	0	1	1	1	1	0	1	1	6
Total	13	12	11	0	1	1	17	17	17	17	0	13	17	8

**Table 3 healthcare-12-00571-t003:** Subgroup analysis regarding HbA1c based on study characteristics.

Characteristics	Subgroup	K	Study ID	N	Overall ES	95% CI		Z (*p*)
						Lower Limit	Upper Limit	
Location (country of publication)	US	6	2,3,6,9,16,17	627	−0.29	−0.45	−0.13	−3.57 (<0.001)
	others	11	1,4,5,7,8,10, 11,12,13,14,15	992	−0.41	−0.82	0.00	−1.98 (0.048)
Participants	<60	7	1,2,5,10, 11,12,15	265	−0.22	−0.46	0.03	−1.74 (0.082)
	≥60	10	3,4,6,7,8,9, 13,14,16,17	1354	−0.46	−0.81	−0.11	−2.56 (0.011)
Study centers	1	5	6,10,11,12,17	321	−0.21	−0.43	0.01	−1.83 (0.067)
	multiple	12	1,2,3,4,5,7,8, 9,13,14,15,16	1298	−0.45	−0.78	−0.12	−2.65 (0.008)
Intervention	r-CGM	6	1,2,5,8,10,30	402	−0.05	−0.25	0.15	−0.49 (0.621)
	rt-CGM	11	3,4,6,7,9,11, 13,14,15,16,17	1217	−0.50	−0.81	−0.18	−3.04 (0.002)
Intervention period (week)	≤24	7	2,5,6,11, 12,16,17	425	−0.24	−0.44	−0.05	−2.47 (0.013)
	>24	10	1,3,4,7,8,9, 10,13,14,15	1194	−0.45	−0.84	−0.07	−2.30 (0.022)
Quality score	≤8	10	1,3,4,6,8,12,13, 14,16,17	1004	−0.31	−0.52	−0.10	−2.94 (0.003)
	>8	7	2,5,7,9,10,11,15	615	−0.43	−0.98	0.12	−1.54 (0.124)
Insulin therapy	Yes	11	1,3,5,7,8,9,10, 11,12,13,14	1188	−0.42	−0.79	−0.05	−2.21 (0.027)
	No	6	2,4,6,15,16,17	431	−0.25	−0.44	−0.05	−2.51 (0.012)

Notes. ES: effect size; CI: confidence interval; r-CGM: retrospective continuous glucose monitoring; rt-CGM: real-time continuous glucose monitoring; and US: The United States.

**Table 4 healthcare-12-00571-t004:** Meta-regression analysis evaluating HbA1c.

Covariate (Ref.)	Estimate	SE	Z	*p*
Location (country of publication; Ref.: others) US	0.26	0.11	2.49	0.013
Participants (Ref.: <60) ≥ 60	−0.28	0.14	−2.06	0.039
Study centers (Ref.: multicenter) one	0.31	0.13	2.47	0.013
Intervention (Ref.: r-CGM) rt-CGM	−0.53	0.12	−4.45	<0.001
Intervention period (Ref.: week > 24) ≤ 24	0.29	0.12	2.49	0.013
Quality assessment (Ref.: >8) ≤ 8	−0.45	0.11	−4.15	<0.001
Receiving insulin therapy (Ref.: not receiving)	−0.29	0.12	−2.49	0.013

Notes. r-CGM: retrospective continuous glucose monitoring; rt-CGM: real-time continuous glucose monitoring; US: The United States; SE: standard error; and Ref.: reference.

**Table 5 healthcare-12-00571-t005:** The effect of CGM intervention on secondary variables.

Variables	Number of Studies	N	Hedge’s G	95% CI		Z (*p*)	I^2^ (%)
				Lower Limit	Upper Limit		
Weight	6 (1,3,6,9,11,17)	593	−0.52	−1.22	0.18	−1.46 (0.145)	93.7
BMI	4 (2,9,11,17)	330	−0.04	−0.27	0.20	−0.30 (0.764)	10.1
Glucose	6 (3,5,8,9,10,13)	688	−0.14	−0.40	0.11	−1.12 (0.263)	56.3
SBP	4 (2,6,9,17)	374	−0.12	−0.34	0.10	−1.09 (0.274)	4.8
DBP	4 (2,6,9,17)	341	0.07	−0.15	0.29	0.63 (0.527)	0
TIR	10 (3,4,5,7,8,9,10,13,15,16)	1110	0.31	−0.14	0.75	1.35 (0.177)	91.3
Hyperglycemia	10 (1,3,4,5,8,9,10,13,15,16)	908	−0.20	−0.49	0.09	−1.35 (0.178)	74.6
Hypoglycemia	10 (1,3,4,5,8,9,10,13,15,16)	898	−0.19	−0.52	0.13	−1.16 (0.246)	79.7
HDL-cholesterol	2 (9,11)	194	−0.33	−0.68	0.03	−1.78 (0.075)	25.9
Distress	3 (3,7,17)	510	−0.08	−0.36	0.20	−0.56 (0.574)	57.8
QoL	3 (3,4,8)	462	−1.29	−3.87	1.29	−0.98 (0.326)	99.2
Satisfaction	3 (8,10,13)	359	2.77	−1.18	6.72	1.38 (0.169)	99.2

Notes. BMI: body mass index; SBP: systolic blood pressure; DBP: diastolic blood pressure; TIR: time in range; QoL: quality of life; HDL: high-density lipoprotein cholesterol; and CI: confidence interval.

**Table 6 healthcare-12-00571-t006:** Publication bias analysis.

Publication Bias Test		Coefficient	SE	95% CI		Z	*p*
				Lower Limit	Upper Limit		
Egger’s regression test	intercept	2.23	1.83	−1.35	5.81	1.22	0.222
	slope	−0.90	0.39	−1.67	−0.14	−2.31	0.021
		tau-b		ties		Z	*p*
Begg’s test	standard	−0.05		4		−0.29	0.771
	corrected	−0.04		4		−0.25	0.805
		Hedge’s		95% CI			
				Lower Limit		Upper Limit	
Trim and fill	original	−0.36		−0.62		−0.10	
	corrected	−0.58		−0.83		−0.33	

Notes. SE: standard error; CI: confidence interval.

## Data Availability

All data generated or analyzed during this study are included in this published article. The datasets used and/or analyzed during the current study are available from the corresponding author on reasonable request.

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
