# Peer review of "Effects of Continuous Glucose Monitoring on Glycemic Control in Type 2 Diabetes: A Systematic Review and Meta-Analysis"

_healthcare, 2024, doi:10.3390/healthcare12050571_

Round 1

Reviewer 1 Report

Comments and Suggestions for Authors

As CGMs are still mostly used by DMT1 patients reviews and studies on the positive effects of CGM on glycemic control of DMT2 patients are necessary and interesting. Although there are already many papers on these effects and there are many countries where CGMs are used by people on DMT2 still reviews like this one is useful. Although the main effects mentioned here is on the reduction of HbA1c there are many others  that could be part of the results by the review (e.g. GV, TIR, dietary quality etc.). The number of studies included in the review could be considered as limited. In general it is a well designed study that clearly conlcudes in the positive effects of CGMs on the glycemic control of DMT2 patients.

  Strengths
  • Although there are many studies published on CGMs and DMT1 there are few on DMT2
  • Most of the studies included were recent (after 2015) and they were conducted in many countries  
  • Tables with the characteristics of the studies included that were well presented and informed and figures were also well presented
  • The effects of CGMs intervention in secondary outcomes is also a strength of the study

weaknesses
  • The total number of studies included (17) is relatively limited compared with the initial number of 491
  • Although the period of time of 12 weeks using the rt-CGM has been considered as a satisfactory period of time in order to get reliable results on the effects of CGM on glycemic control a longer period of time > 12 weeks is recommended
  • The lack of information on education during intervention

Author Response

We appreciate the time and effort that you and the reviewers have put into providing valuable feedback and insightful comments, which improved our manuscript. We have carefully considered each comment and updated the manuscript, as required. We have marked the revisions made to the manuscript in red font.

Reviewer 2 Report

Comments and Suggestions for Authors

I am honored to serve as a reviewer on a manuscript by Kong S-Y and Cho M-K that reports the results of a systematic review and meta-analysis of studies evaluating the effects of continuous glucose monitoring (CGM) on glycemic control in patients with type 2 diabetes. The study was carried out at a high methodological level following modern standards estimated for such research. A subanalysis of the included studies allowed the authors to identify differences in the effect of CGM depending on patient age, duration of CGM use, and other characteristics. The results have significant implications for clinical practice. I highly recommend this manuscript for publication.

At the same time, I have two suggestions for the authors.

1.     Following the selected outcome, the authors analyze the effect of CGM on HbA1c levels. However, HbA1c is not the only indicator of glycemic control. The rate of hypoglycemia and the time in range were not taken into account. This should be stated as a limitation of the study or additional information provided.

2.     CGM may have different effects on glycemic control in patients managed with insulin and those not receiving insulin therapy. This meta-analysis included studies with different antihyperglycemic treatment modalities. Were studies sub-analyzed based on insulin use? If yes, please provide the results. I think this topic deserves mention in the Discussion.

Author Response

(The authors gave the same response as above.)

Reviewer 3 Report

Comments and Suggestions for Authors

With the increased interest in reversing type 2 diabetes and the availability of the technology, this is an important review.

However, there are areas where this review can be enhanced. 

Methodology:  while the methodology appears sound, there are practical challenges that may further review given HbA1c was the primary outcome measure. 

Firstly, there was no distinction between CGM being introduced subsequent to stable medications, or both groups receiving upgraded care (e.g. commencing insulin therapy as per Blackberry 2014) and the intervention group receiving CGM as well.  Although CGMs may better prevent hypoglycaemia episodes compared to SMBG, increased time spent in the hypoglycaemia range will lower HbA1c in a way that does not reflect better glycaemic control. 

With the longer studies, there is no discussion about the frequency of use of the CGM.  For example, both the Ehrhardt and Furler studies used intermittent periods of CGM use, but the periods were different. 

Please check the data, unclear why Furler is reported in the table as 48 weeks, when the paper reports a 52  week study. 

Please be aware of terminology - devices such as Dexcom are commonly known as 'real-time' CGM (or rtCGM) as they continuously update the reading device.  While the Libre were considered 'flash' CGM  (fCGM) as the sensor had to be scanned at least every 8 hours to download the data to the reader.  Both devices could be used for 'retrospective' or 'real-time' monitoring depending on whether the participant had instant access to their data or had to wait for their clinician to give then their results.   This is an important confounder for the use of CGMs. 

It is unclear why the Allen study is included as the CGM was only worn for three days and an 8 week study should not have HbA1c as a primary outcome. 

There is limited discussion about the introduction or cessation of other medications, and their impact on HbA1c.  This should be considered a secondary measure, as a person could have significant changes to their medications, but this may not be reflected in the HbA1c. 

Results - no significant comments

Introduction:  The figures presented are from a Japanese study a population with generally fewer diabetes related complications.  They also only mention the cardiovascular complications.  For greater impact, you may want to consider overall life expectancy and/or other comorbidities.  for example,   10.1016/S2213-8587(23)00223-1

Discussion.  Please consider that most of these studies are 12 weeks in length as this is generally considered the minimum practical period if you are using HbA1c as a primary outcome measure. 

The conclusion reached about CGM in office workers (lines 302-308) is a non-sequitur.  CGMs are easily used by office workers, so this is not the reason for their self-management challenges or poor blood glucose control.  Please consider this review on young-onset T2D  https://www.sciencedirect.com/science/article/abs/pii/S2213858717301869

Comments on the Quality of English Language

Overall, language quality is good.  Please review lines 31 to 35 as it is a long sentence, but also please see suggestion above for greater impact.  . 

Author Response

(The authors gave the same response as above.)

Round 2

Reviewer 3 Report

Comments and Suggestions for Authors

Thank you for addressing the suggestions and recommendations from the previous review. 

Comments on the Quality of English Language

There is still the occasional sentence that would benefit from better English grammar  - e.g. from approx line 320. 

"Therefore, it is required the development of an efficient way to manage blood sugar levels using CGM, which can be used by office workers without restrictions on time and place."  

Author Response

(The authors gave the same response as above.)
